# Identification, Culture Characteristics and Whole-Genome Analysis of *Pestalotiopsis neglecta* Causing Black Spot Blight of *Pinus sylvestris* var. *mongolica*

**DOI:** 10.3390/jof9050564

**Published:** 2023-05-12

**Authors:** Jing Yang, Shuren Wang, Yundi Zhang, Yunze Chen, Heying Zhou, Guocai Zhang

**Affiliations:** 1College of Forestry, Guizhou University, Huaxi District, Guiyang 550025, China; zhydyx2019@163.com; 2Heilongjiang Province Key Laboratory of Forest Protection, School of Forest, Northeast Forestry University, Hexing Road 26, Xiangfang District, Harbin 150040, China; wang02102022@126.com (S.W.); 13251611896@163.com (Y.Z.); 3School of Biological Sciences, Guizhou Education University, Wudang District, Guiyang 550018, China; chenyunze@gznc.edu.cn

**Keywords:** fungal genetics, gene annotation, *Pestalotiopsis neglecta*, phytopathogenic fungus

## Abstract

Black spot needle blight is a serious conifer disease of *Pinus sylvestris* var. *mongolica* occurring in Northeast China, which is usually caused by the plant pathogenic fungus *Pestalotiopsis neglecta*. From the diseased pine needles collected in Honghuaerji, the *P. neglecta* strain YJ-3 was isolated and identified as the phytopathogen, and its culture characteristics were studied. Then, we generated a highly contiguous 48.36-Mbp genome assembly (N50 = 6.62 Mbp) of the *P. neglecta* strain YJ-3 by combining the PacBio RS II Single Molecule Real Time (SMRT) and Illumina HiSeq X Ten sequencing platforms. The results showed that a total of 13,667 protein-coding genes were predicted and annotated using multiple bioinformatics databases. The genome assembly and annotation resource reported here will be useful for the study of fungal infection mechanisms and pathogen–host interaction.

## 1. Introduction

*Pinus sylvestris* var. *mongolica* Litv. (Pinales: Pinaceae), a variety of European *P. sylvestris* L., is one of the excellent trees for soil and water conservation in Northeast China [1]. In recent years, coniferous diseases of *P. sylvestris* var. *mongolica* occurred frequently in the Honghua’erji area, Inner Mongolia. In addition to the common blight caused by *Sphaeropsis sapinea*, black spot needle blight is also a serious conifer disease there, which can affect growth and production [2]. This disease first appeared on the upper part of the needles, and the needles then became withered and gradually showed light black spots, although they remained green. As the disease progressed, the needles eventually died and turned gray with many dark black spots. It was reported that there are two dominant pathogens causing black spot blight of *P. sylvestris* var. *mongolica* in China: *Pestalotiopsis neglecta* and *Heterotruncatella spartii* [2,3]. However, *P. neglecta* has not been found in the Honghua’erji area before.

*Pestalotiopsis* Steyaert is an appendage-bearing conidial anamorphic form (coelomycetes) in the family Amphisphaeriaceae that includes more than 225 species [4,5,6,7,8]. The genus has received much attention from the scientific community, because recent reports confirmed that *Pestalotiopsis* spp. are not only common phytopathogens that cause a variety of diseases, but are also endophytes that possess diverse metabolic capabilities, including the production of potentially valuable secondary metabolites [9,10]. As a plant pathogen, *Pestalotiopsis* is known mostly from the tropics, where it causes leaf blights in many plant species [11,12,13]. Species may also cause rot of fruit and other postharvest diseases [12,14,15]. Besides black spot blight of *P. sylvestris* var. *mongolica*, *P. neglecta* can also cause chlorotic spot in maize (*Zea mays*), shoot blight in *Cryptomeria japonica*, and canker disease in blueberries (*Vaccinium corymbosum*) [8,16,17].

In addition to conventional culture characteristics research, high-throughput whole-genome sequencing is an effective method to understand the relevant characteristics of strains more comprehensively at the gene level. Until now, some strains of *Pestalotiopsis* spp. (such as *P. fici*) have been whole-genome sequenced [18], while the genome assembly of a pathogenic *Pestalotiopsis* sp. has not yet been released. The culture characteristics and genome assembly of the pathogenic fungus *Pestalotiopsis neglecta* strain YJ-3 obtained in this study will provide support for the systematic analysis of the pathogenicity mechanism and interaction mechanism with the host at the molecular level.

## 2. Materials and Methods

### 2.1. Isolation and Identification of P. neglecta

Black needle blight symptom conifers were collected from *P. sylvestris* var. *mongolica* forest in the Honghua’erji Forest Farm in August 2020. Infected needles were surface sterilized with 75% ethanol and rinsed in sterile deionized water, after which they were plated on potato dextrose agar (PDA) and incubated at 25 °C for 48 h. Mycelia were then subcultured on new PDA plates for isolation and purification. Mycelial colonies were observed after 7 days, and conidia were examined under a light microscope. Fungal genomic DNA was extracted from mycelia on PDA plates using a DNA extraction kit (Sangon). Species names for this isolate were determined using the primer pair ITS1 (5′-TCCGTAGGTGAACCTGCGG-3′) and ITS4 (5′-TCCTCCGCTTATTGATATGC-3′) [19]. The ITS sequence is deposited in NCBI GenBank with accession number OQ691957.

Healthy young pine needles were collected for Koch’s postulates verification [3]. The sporulated *P. neglecta* mycelial plugs (3 mm in diameter) were excised from a 12-day-old PDA plate and placed on wounded pine needles to initiate infection. Empty plugs from sterile PDA plates were used as controls. Infected pine needles were placed in Petri dishes (150 mm in diameter) with moisture filter paper and incubated at 25 °C with a 12 h photoperiod. Symptom development was observed daily.

### 2.2. Culture Characteristics of P. neglecta

#### 2.2.1. Screening of Optimal Medium

Potato dextrose agar (PDA) medium, Czapek–Dox agar medium (CDA) (sodium nitrate 3 g, dipotassium hydrogen phosphate 1 g, magnesium sulfate 0.5 g, potassium chloride 0.5 g, ferrous sulfate 0.01 g, sucrose 30 g, agar 20 g, and distilled water to complete 1000 mL) [20,21], Sabouraud dextrose agar medium (SDA) [22] and yeast extract peptone sugar agar medium (YEPSA) were used to study the effect of different media on mycelial growth of strain YJ-3. The mycelial plug (5 mm in diameter) was taken from the edge of the activated strain colony and inoculated in the center of the above-mentioned medium plate, respectively. The colony diameter in each plate was measured after culturing for 7 d at 25 °C under the natural light conditions.

#### 2.2.2. Screening of Carbon and Nitrogen Sources

PDA medium was used as the basic medium. The 30 g glucose was replaced with 30 g mannitol, glycerol, sucrose, maltose, soluble starch or lactose to test these compounds as carbon sources. The 3 g yeast extract, peptone, tryptone, ammonium nitrate or ammonium sulfate were added to determine growth on different nitrogen sources. A 7-day-old mycelial plug (in 5 mm diameter) was transferred to the center of each sole carbon source medium and sole nitrogen source medium. Colony growth was determined by measuring the colony diameters after incubation for 5 d at 25 °C under the natural light conditions [23].

#### 2.2.3. Temperature and pH Tests

Strain YJ-3 was used to evaluate the effects of temperature and pH on colony growth on PDA plates. Temperatures were tested at 20, 22, 24, 25, 26 and 28 °C. 

To clarify the effect of pH on radial mycelial growth, PDA media were adjusted with 0.1 M NaOH and 0.1 M HCl to obtain pH values from 5.0 to 12.0 at a pH 1.0 interval (i.e., 5.0, 6.0, 7.0, 8.0, 9.0, 10.0, 11.0, and 12.0). A 5 mm-diameter mycelial plug was placed in the center of a 90 mm petri dish with PDA medium and incubated at 25 °C in the dark, with three replicates for each treatment [24]. 

The effects of pH and temperature on mycelial growth were determined by measuring the colony diameters after 5 d of incubation. Data were analyzed in IBM SPSS Statistics (V22.0, IBM Corp., Armonk, NY, USA) to select the model that best fit the individual data points, and SPSS was used to confirm the selected model. The optimal temperature and pH value of the regression curves were calculated based on the regression equations generated by IBM SPSS Statistics.

### 2.3. Whole-Genome Sequencing and Assembly

#### 2.3.1. DNA Extraction

Genomic DNA was extracted using the Omega Fungal DNA Kit D3390-02 according to the manufacturer’s protocol. Purified genomic DNA was quantified by TBS-380 fluorometer (Turner BioSystems Inc., Sunnyvale, CA, USA). High quality DNA (OD260/280 = 1.8~2.0, >15 μg) was used to further construct the library.

#### 2.3.2. Genome Sequencing and Assembly

The extracted genomic DNA was sequenced using a combination of the PacBio RS II Single Molecule Real Time (SMRT) and the Illumina HiSeq X Ten sequencing platforms (Shanghai Majorbio Bio-Pharm Technology Co., Ltd., Shanghai, China). The Illumina data were used to assess genome size, heterozygosity, duplication and presence of contamination to aid in the selection of subsequent assembly strategies.

For Illumina sequencing, at least 5 μg genomic DNA samples were used in sequencing library construction. The DNA samples were sheared into 400–500 bp fragments using a Covaris M220 Focused Acoustic Shearer following the manufacturer’s protocol. Illumina sequencing libraries were then prepared from these fragments using the NEXTflex™ Rapid DNA-Seq Kit. Briefly speaking, 5’ prime ends were first end-repaired and phosphorylated. Next, the 3’ ends were A-tailed and ligated to sequencing adapters. The third step was to enrich the adapter-ligated products using PCR. The prepared libraries were then used for paired-end Illumina sequencing (2 × 150 bp) on an Illumina HiSeq X Ten platform.

For Pacific Biosciences sequencing, an aliquot of 8 μg DNA was spun in a Covaris g-TUBE (Covaris, Woburn, MA, USA) at 6000 r/min for 60 s using an Eppendorf 5424 centrifuge (Eppendorf, Westbury, NY, USA). DNA fragments were purified, end-repaired and ligated with SMRTbell sequencing adapters according to the manufacturer’s recommendations (Pacific Biosciences, Menlo Park, CA, USA). The resulting sequencing library was then purified three times using 0.45 × volumes of Agencourt AMPure XP beads (Beckman Coulter Genomics, Danvers, MA, USA). Finally, a ~10kb insert library was prepared and sequenced on one SMRT cell using standard methods.

The complete genome sequence was assembled using both the PacBio reads (>80×) and Illumina reads (>100×). The original image data were transferred into sequence data via base calling, which is defined as raw data or raw reads. A statistic of quality information was applied for quality trimming, by which the low-quality data can be removed to form clean data (FastqTotalHighQualityBase.jar): (1) The adapter sequence was removed from the reads; (2) The bases containing non-A, -G, -C, and -T at the 5′ end were cut and removed; (3) The read ends with lower sequencing quality (sequencing quality value of less than Q_20_) were trimmed; (4) The reads containing up to 10% N were removed; (5) Small fragments with length of less than 25 bp were discarded after removing the adapter and quality trimming.

The clean reads were then assembled into a contig using SOAPdenovo v2.04, canu v1.7 and Flye v2.8.3 [25]. The last circular step was checked and finished manually, generating a complete genome with seamless chromosomes and plasmids. Finally, the error correction of the PacBio assembly results was performed using the Illumina reads with Pilon v1.23 [26].

#### 2.3.3. Gene Prediction and Functional Annotation

The data generated from PacBio and Illumina platform were used for bioinformatics analysis. All of the analyses were performed using the free online platform of Majorbio Cloud Platform (cloud.majorbio.com, accessed on 26 March 2023) from Shanghai Majorbio Bio-pharm Technology Co., Ltd. The detailed procedures are as follows.

Genomic structural analysis was performed by promoter prediction and genome mapping to comprehensively grasp the genome of the *P. neglecta* strain YJ-3. MAKER v2.31.9 [27], tRNA-scan-SE v1.3.1 [28] and Barrnap v0.4.2 were used for CDS, tRNA and rRNA prediction, respectively. For functional annotation, BLASTP was used to search against the SWISS-PROT and nonredundant (NR) protein databases of NCBI. The protein homology search in Diamond v0.8.35 was used to assign the gene ontology (GO) terms, Kyoto Encyclopedia of Genes and Genomes (KEGG) pathways and eukaryotic cluster information to a certain protein [29]. For protein annotation, the eggNOG database [30] was searched using HMMER v3.1b2 [31]. Secreted proteins were predicted using SignalP v4.1 [32], and Carbohydrate-Active Enzymes were detected using the CAZymes Analysis Toolkit v6.0 [33]. The Pathogen–Host Interaction database v4.4 [34] was used to predict potential pathogenically active proteins.

### 2.4. Data Availability

This Whole Genome Shotgun project was deposited at DDBJ/ENA/GenBank under the accession JAQJCZ000000000 (BioProject PRJNA925307; BioSample SAMN32787139). The version described in this paper is version JAQJCZ010000000.

## 3. Results

### 3.1. Isolation and Identification of P. neglecta

*Piuns sylvestris* var. *mongolica* needles infected with black spot needle blight are often found in summer (Figure 1A). To determine the pathogen causing this infection, diseased needles were collected on the Honghuaerji Forest Farm and cultured in the PDA medium. The amplified ITS sequence (OQ691957) was aligned by NCBI BLASTn, which showed that the strain YJ-3 was identified as *P. neglecta*. The fungal colony was round with smooth edges. It was white aerial hypha in the early stage, and then slowly turned into light yellow (Figure 1B). Black pycnidia were irregularly distributed in mature colonies after 10 d. Microscopic observation showed that the conidia were five-celled clavate spindles. The color of the middle three cells was generally grayish brown to dark brown, and the fourth cell was slightly lighter in color and slightly constricted at the septum (Figure 1C). We sprayed the spore suspension of the isolated YJ-3 on the healthy needles, and after 15 d, we observed the protruding black spots on the needles (Figure 1D). These results showed that isolated *P. neglecta* YJ-3 was the pathogen causing black spot needle blight of *P. sylvestris* var. *mongolica*.

### 3.2. Culture Characteristics of P. neglecta YJ-3

The effects of different medium, carbon sources and nitrogen sources on the growth of *P. neglecta* YJ-3 reached a significant level (Figure 2). Different media significantly affected the growth of the strain. The growth rates of the strain YJ-3 on PDA and SDA were significantly higher than those of others (*p* < 0.05), and the growth rates were relatively slower on YEPSA and CDA (Figure 2A). The colony diameter of the strain YJ-3 in the medium with glucose, maltose and sucrose as the carbon source were significantly larger than those of other carbon sources (Figure 2B), and the growth in the medium with tryptone, ammonium nitrate and ammonium sulfate as the nitrogen source were significantly better than that in other treatments, indicating that the growth of the strain YJ-3 might require higher nitrogen content (Figure 2C). Thus, PDA and SDA are the most suitable media, glucose and maltose are the most suitable carbon sources, and tryptone, ammonium nitrate and ammonium sulfate are the most suitable nitrogen sources for the growth of *P. neglecta* YJ-3.

The effect of different culture temperature and pH on the growth of strain YJ-3 is shown in Figure 2D,E. The mycelia of PYJ-3 grew normally between 20 and 28 °C. After 7 days of culture, the diameter of the colony growing at 25 °C was the largest, reaching 8.13 ± 0.15 cm. As shown in Figure 2E, YJ-3 did not have strict requirements on the pH of the culture medium. The mycelia grew well within the range of pH 5 to 12, of which pH 5 to 9 were more suitable for mycelia growth, and especially at pH 7, the colony diameter reached the largest size, up to 8.32 ± 0.12 cm.

### 3.3. Genome Sequencing and Assembly

After filtering the low-quality and short reads, the final assembled genome of *P. neglecta* YJ-3 was 48.36 Mb and consisted of eight scaffolds in total, with the longest scaffold length of 8.96 Mb, an average length of 6.04 Mbp, and an N50 length of 6.62 Mbp (Table 1). 

The genome completeness was assessed with BUSCO (benchmarking universal single-copy ortholog genes) v3.0 using the Ascomycota dataset, resulting in a coverage rate of 98.6%. The overall G + C content assembly was 51.73%. The results showed that repetitive sequences represented 0.06% of the genome (Table 1). However, the actual repeat content is expected to be larger as small contigs shorter than 1 kb (we filtered them out) may contain many repetitive sequences.

### 3.4. Gene Prediction and Functional Annotation

In the present study, it was predicted that the *P. neglecta* strain YJ-3 has 13,667 genes found in the genome, accounting for 54.74 % of the total genome length, and the average length of each gene is 1936.62 bp. 

Promoters are regulatory elements that regulate gene expression and determine the intensity and timing of gene expression. The insertion or deletion of the promoter can change the expression mode of the gene and realize the research on the regulation of the growth and metabolism of the cell. PromPredict software was used to predict and analyze the promoter sequence of the gene, and the promoters of 8334 genes were identified (Table 1).

Moreover, 13,667 protein-coding genes were annotated using several databases. Among them, 12,764 (94.32%) genes were annotated in the NCBI nr database, followed by the eggNOG database (11,396, 83.39%), the Go database annotations (9180, 67.17%), the Swiss–Prot database (9020, 66.00%), and the KEGG database (4112, 30.09%). In addition, for non-coding RNAs, the *P. neglecta* strain YJ-3 had 238 tRNAs and 46 rRNAs.

The NCBI nr database is a non-redundant protein database. In this study, 12,764 genes were annotated in NCBI nr and the information of the 58 genes whose sequence identity was 100% is listed in Appendix A. 

The eggNOG database was used to annotate the function of the *P. neglecta* YJ-3 protein. The cluster analysis results of 3185 genes annotated are shown in Figure 3. There were 5806 genes that did not have clear functions, which may be related to the lack of genomic research of *P. neglecta* and lack of reference genes. The most annotated genes with clear functional classification were carbohydrate transport and metabolism (917 genes). The genes related to amino acid transport and metabolism, lipid transport and metabolism, nucleotide transport and metabolism were 535, 337 and 131, respectively. In addition, there were 530 genes related to the biosynthesis, transport and catabolism of secondary metabolites, and 110 genes related to cell wall/membrane/envelope biogenesis.

The GO database classifies gene functions into molecular functions, cell components and biological processes, with a gene annotated multiple times through the GO item. The results of the *P. neglecta* YJ-3 genome annotated by the GO database are shown in Figure 4, and there were 7279 genes annotated for molecular function, 4957 genes annotated for cell component and 3467 genes annotated for biological process.

Furthermore, KEGG database is a large knowledge base for the systematic analysis of gene function, as well as connection of genomic information and functional information. The results showed that a total of 4112 genes were annotated, divided into 6 major categories and 46 sub-categories, as shown in Figure 5.

### 3.5. Carbohydrate-Active Enzymes (CAZymes)

Carbohydrate-active enzymes (CAZymes) play important roles in the breakdown of complex carbohydrates and for phytopathogenic fungi, and some kinds of CAZymes are responsible for the acquisition of nutrients from plants as well as works in the process of infection and colonization [35,36]. As shown in Figure 6A, a total of 798 putative CAZyme genes were identified in the *P. neglecta* strain YJ-3, including 327 glycoside hydrolases (GHs), 73 glycosyl transferases (GTs), 224 auxiliary activities (AAs), 142 carbohydrate esterases (CEs), 26 polysaccharide lyases (PLs), and 5 carbohydrate-binding modules (CBMs).

### 3.6. Pathogenic System Analysis

Secondary metabolites, especially fungal toxins, are believed to be involved in the pathogenicity of many phytopathogenic fungal species and can be described as potential virulence factors [37,38]. Diamond software version 0.8.35 was used to compare the putative proteins with the database of functional virus factors (DFVF) to analyze the virulence-related genes in the *P. neglecta* strain YJ-3. The results showed that 1439 genes were identified as fungal virulence factors in total (Table 1).

The Pathogen–Host Interaction database [34] was used to predict potential pathogenically active proteins. Shown in Figure 6B, 2036 genes were predicted to play roles in pathogen–host interactions, including 149 genes that were identified as increased pathogenicity (hypervirulence). Fifty-eight genes with a single description of increased pathogenicity (hypervirulence) are listed in Table 2, which might be the key pathogenic genes of the *P. neglecta* strain YJ-3.

Moreover, antibiotic resistance genes were annotated through the Comprehensive Antibiotic Resistance Database (CARD), which is constructed in the form of Antibiotic Resistance Ontology (ARO) as a taxonomic unit, aiming to associate antibiotic modules and their targets, resistance mechanisms, gene mutations and other information. In the present study, a total of 175 genes were annotated in the CARD database (Figure 6C), with the top three drug class tetracycline antibiotics (75 genes), fluoroquinolone antibiotics (28 genes) and penam (25 genes).

### 3.7. Cytochromes P450

Cytochromes P450 (CYP450) is a large family of proteins with ferroheme as a cofactor. They can catalyze the oxidation reaction of many substrates and participate in the metabolism of endogenous substances and exogenous substances including drugs and environmental compounds. In the genome of the *P. neglecta* strain YJ-3, a total of 368 CYP450 genes were annotated. Among them, 46 genes were identified as proteins from the CYP51 subfamily (Appendix A).

## 4. Discussion

*Pinus sylvestris* var. *mongolica* is widely planted in China as a windbreak and sand fixation tree [38]. In this study, the *P. neglecta* strain YJ-3 causing black spot needle blight was isolated and identified from the infected needles of *P. sylvestris* var. *mongolica*, and its culture characteristics were studied. In addition, the whole genome of the *P. neglecta* strain YJ-3 was sequenced, assembled and annotated using a combination of PacBio RS II Single Molecule Real Time (SMRT) and the Illumina HiSeq X Ten sequencing platforms. 

In our study, the genome size and GC content of the *P. neglecta* strain YJ-3 was 48.36 Mb and 51.73%, respectively. It was reported that the genome size of *P.fici* W106-1 was 52 Mb and the GC content was 48.73% [18]. The proportion of repeat sequences in the *P. neglecta* strain YJ-3 (0.06%) is smaller than that in the *P. fici* (2.97%). The lower proportion of repeat sequences may be the reason why the genome size of the *P. neglecta* strain YJ-3 was smaller than that of the *P. fici* W106-1.

Glycoside hydrolase (GH) enzymes have the potential to hydrolyze complex carbohydrates, and glycosyltransferase (GTs) are important for surface structures recognized by the host immune system [39]. The prediction of CAZymes in the *P. neglecta* strain YJ-3 genome contains 327 GHs genes, 73 GTs genes, and 26 polysaccharide lyases (PLs). High amounts of GTs and GHs indicate that this strain has a high potential to destroy plant cell walls during infection [40].

*Pestalotiopsis neglecta* strain YJ-3 has been proven to infect the needles of *P. sylvestris* var. *mongolica* and induce the occurrence of black spot needle blight [3]. Through annotation of the genome of the *P. neglecta* strain YJ-3, we found that this strain has 2036 genes that were predicted to play roles in pathogen–host interactions, providing convenience for understanding its infection mechanism in the future [41].

In addition, CYP is involved in many important cellular processes, such as the conversion of hydrophobic intermediates in primary and secondary metabolic pathways and the detoxification of natural and environmental pollutants [42]. Among them, given the importance and specificity of ergosterol in fungal cells, the key enzyme in its biosynthetic pathway 14 α- demethylase (CYP51) has become an ideal target for antifungal drugs [43]. In the genome of the *P. neglecta* strain YJ-3, a total of 368 CYP450 genes were annotated and 46 genes were identified as proteins from the CYP51 subfamily. The identification of these genes provides a research basis for the subsequent development of specific fungicides against *P. neglecta*.

To conclude, in the present study, a strain of *P. neglecta* named YJ-3 was isolated and identified from the infected needles of *P. sylvestris* var. *mongolica* with black spot needle blight, and the culture characteristics were studied. Moreover, the draft genome sequence assembly and bioinformatics analysis of the *P. neglecta* strain YJ-3 were provided here, which represents a useful source for future research on fungal comparative genomics studies, infection mechanisms and pathogen–host interaction.

## Figures and Tables

**Figure 1 jof-09-00564-f001:**
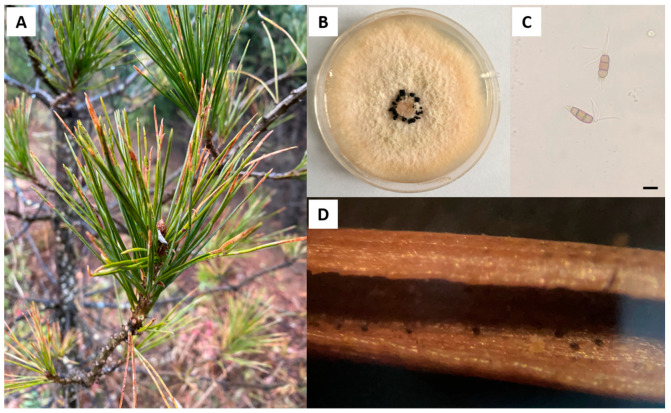
*Pestalotiopsis neglecta* YJ-3 causing black spot needle blight on *Pinus sylvestris* var. *mongolica*. (**A**) Infected diseased pine trees; (**B**) Colony surface on PDA medium; (**C**) Conidia; (**D**) Disease symptoms. Scale bars = 10 μm.

**Figure 2 jof-09-00564-f002:**
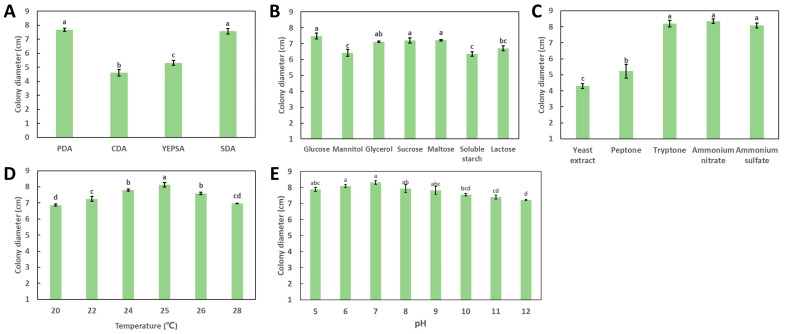
Culture characteristics of the *P. neglecta* strain YJ-3. (**A**–**C**): The effects of different mediums (**A**), carbon sources (**B**) and nitrogen sources (**C**) on mycelial growth of the *P. neglecta* strain YJ-3. (**D**,**E**): The effects of different temperatures (**D**) and pH (**E**) on mycelial growth of the *P. neglecta* strain YJ-3. Letters (a, b, c, d) represent statistically significant differences between different conditions (*p* < 0.05).

**Figure 3 jof-09-00564-f003:**
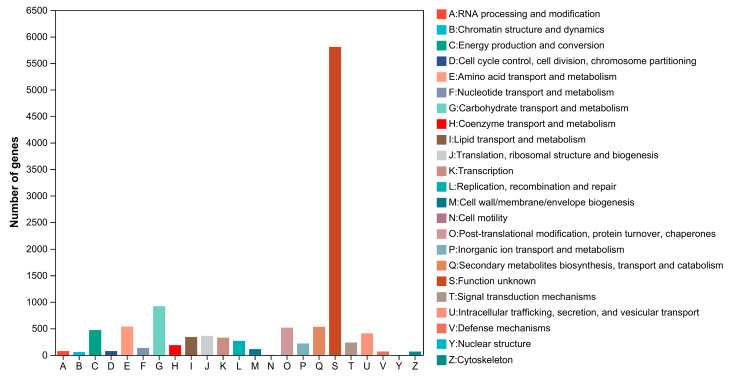
The eggNOG functional classification diagram.

**Figure 4 jof-09-00564-f004:**
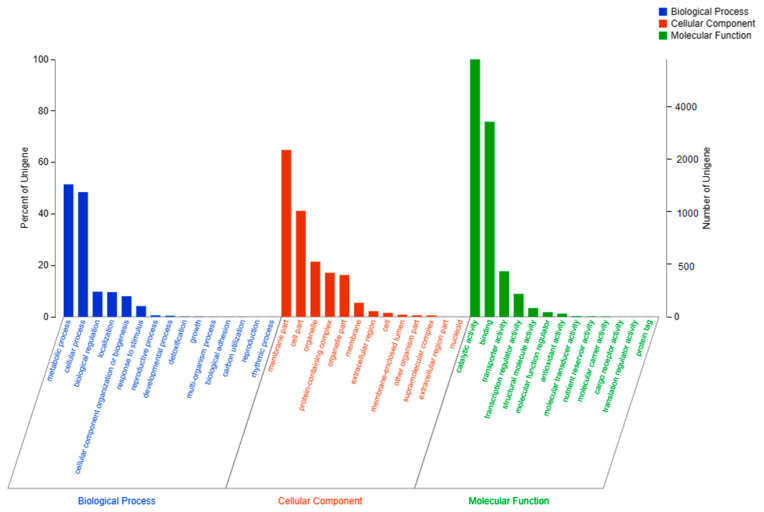
GO annotation of the *P. neglecta* strain YJ-3 genome.

**Figure 5 jof-09-00564-f005:**
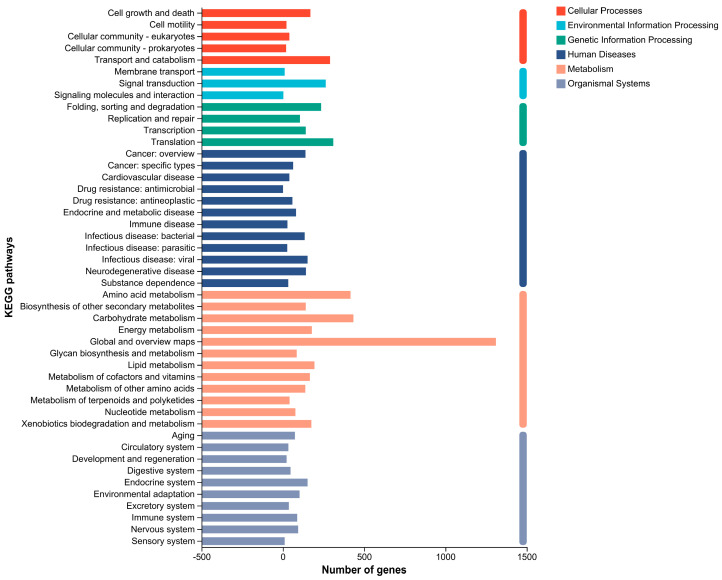
KEGG classification of the *P. neglecta* strain YJ-3 genome.

**Figure 6 jof-09-00564-f006:**
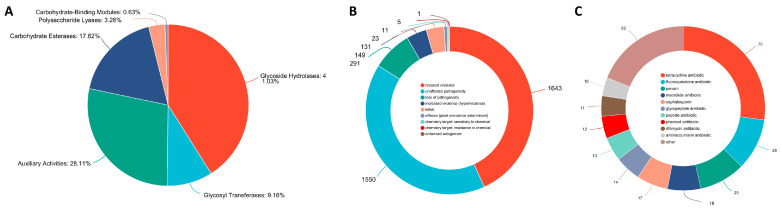
Predicted carbohydrate active enzymes (**A**), pathogenically active proteins (**B**) and antibiotic-resistance genes (**C**) of *Pestalotiopsis neglecta* strain YJ-3.

**Table 1 jof-09-00564-t001:** Genome characteristics of the *P. neglecta* strain YJ-3.

Features		Values
Reads features (PacBio)	Total read number	168,409
	Total bases (Mbp)	3449.18
	Average read length (Kbp)	20.48
	Largest read length (Kbp)	55.65
Genome features	Predicted genome size (Mbp)	48.36
	BUSCOs (%)	98.6
	GC content (%)	51.73
	Contig number	8
	Contig N50 (Mbp)	6.62
	Contig N90 (Mbp)	5.65
	Average contig length (Mbp)	6.04
	Largest contig length (Mbp)	8.92
	Repeat sequence (%)	0.06
	Protein-coding genes	13,667
	Gene Density (number of genes per Kbp)	0.28
	tRNA genes	238
	Candidate secreted proteins	1580
	Carbohydrate-active enzymes	798
	Pathogen host interactive genes	2036
	Fungal virulence factors	1439

**Table 2 jof-09-00564-t002:** Fifty-eight increased virulence (hypervirulence) genes in the *P. neglecta* strain YJ-3 genome.

Gene ID	Location	PHI ID	Protein ID	Pathogen Species	Function	Identity (%)
gene00540	Scaffold1	PHI:4918	B4EIW7	*Burkholderia cenocepacia*	Global virulence regulator	33.3
gene00727	Scaffold1	PHI:7014	Q63KU5	*Burkholderia pseudomallei*	Putative peptide synthase/polyketide synthase; proteasome inhibitor	23.9
gene00747	Scaffold1	PHI:6957	A0A0J9WX00	*Xanthomonas oryzae*	Dehydrogenase	25.6
gene01771	Scaffold1	PHI:3607	K7NCV2	*Epichloe festucae*	non-ribosomal peptide synthetase Gene (sidN) encoding a siderophore synthetase	35
gene01838	Scaffold1	PHI:4613	D0ZDL7	*Edwardsiella tarda*	Type III Secretion System	24.6
gene02362	Scaffold1	PHI:3630	L7N655	*Mycobacterium tuberculosis*	Probable aminopeptidase	34.1
gene02365	Scaffold1	PHI:3793	Q6FM27	*Candida glabrata*	Encode alpha-(1-6)-mannosyl-transferases	42.6
gene02548	Scaffold1	PHI:2393	I1R980	*Fusarium graminearum*	-	26.8
gene02685	Scaffold2	PHI:2393	I1R980	*Fusarium graminearum*	-	32.4
gene02746	Scaffold2	PHI:7014	Q63KU5	*Burkholderia pseudomallei*	Putative peptide synthase/polyketide synthase; proteasome inhibitor	33.6
gene02944	Scaffold2	PHI:2393	I1R980	*Fusarium graminearum*	-	28.7
gene03052	Scaffold2	PHI:2313	F8R4 × 8	*Metarhizium anisopliae*	Dihydroxynaphthalene Melanin Biosynthesis	67.1
gene03080	Scaffold2	PHI:5268	C8V0N8	*Aspergillus nidulans*	Epimerase	66.1
gene03215	Scaffold2	PHI:2393	I1R980	*Fusarium graminearum*	-	39.2
gene03814	Scaffold2	PHI:7048	Q5A663	*Candida albicans*	Intramitochondrial quality control protease	31.9
gene03868	Scaffold2	PHI:2393	I1R980	*Fusarium graminearum*	-	25.8
gene03896	Scaffold2	PHI:3529	J9VQL2	*Cryptococcus neoformans*	Phosphate Acquisition and Storage	46.9
gene04733	Scaffold3	PHI:2393	I1R980	*Fusarium graminearum*	-	26.8
gene05524	Scaffold3	PHI:7121	B0YCA2	*Aspergillus fumigatus*	Trehalose–phosphate synthase subunit	51.9
gene05651	Scaffold3	PHI:2393	I1R980	*Fusarium graminearum*	-	26
gene05825	Scaffold3	PHI:9218	B8NRT5	*Aspergillus flavus*	Peptidyl–prolyl cis–trans isomerase	40.4
gene05939	Scaffold3	PHI:9236	Q18CB9	*Clostridioides difficile*	Cyclophilin-type peptidyl–prolyl-cis/trans–isomerase	36
gene06010	Scaffold3	PHI:9225	A0A1D8PCN8	*Candida albicans*	Phosphoprotein phosphatase PP4 regulatory subunit	32.3
gene06177	Scaffold3	PHI:9226	A0A1D8PSJ8	*Candida albicans*	Phosphoprotein phosphatase PP4 catalytic subunit	68.2
gene06187	Scaffold3	PHI:4468	A0A068BFA5	*Epichloe festucae*	Cell-Wall Integrity MAPK	30.1
gene06309	Scaffold3	PHI:7619	Q8Y755	*Listeria monocytogenes*	DExD-box RNA-helicase	31.3
gene06574	Scaffold3	PHI:3630	L7N655	*Mycobacterium tuberculosis*	Probable aminopeptidase	30.2
gene06896	Scaffold4	PHI:6751	Q4WQ36	*Aspergillus fumigatus*	Initiates asexual development	38.1
gene07319	Scaffold4	PHI:3085	P42375	*Porphyromonas gingivalis*	folding of newly synthesized proteins, preventing misfolding and aggregation	53.6
gene07390	Scaffold4	PHI:6548	F9WZ47	*Zymoseptoria tritici*	Involved in virulence and host-specific disease development	32
gene07781	Scaffold4	PHI:3630	L7N655	*Mycobacterium tuberculosis*	Probable aminopeptidase	38.5
gene08414	Scaffold4	PHI:494	Q4WPX2	*Aspergillus fumigatus*	Fatty acid oxygenase	34.1
gene08619	Scaffold4	PHI:5267	E9R863	*Aspergillus nidulans*	Epimerase	52
gene08738	Scaffold5	PHI:6956	A0A0J9WWZ9	*Xanthomonas oryzae*	3-oxoacyl-ACP reductase	28.7
gene08982	Scaffold5	PHI:6956	A0A0J9WWZ9	*Xanthomonas oryzae*	3-oxoacyl-ACP reductase	28.2
gene09589	Scaffold5	PHI:6751	Q4WQ36	*Aspergillus fumigatus*	Initiates asexual development	29
gene09837	Scaffold5	PHI:2393	I1R980	*Fusarium graminearum*	-	28.9
gene09987	Scaffold5	PHI:672	P0CM56	*Cryptococcus neoformans*	Capsule polysaccharide biosynthesis	32.1
gene10593	Scaffold6	PHI:2393	I1R980	*Fusarium graminearum*	-	28.9
gene10726	Scaffold6	PHI:6644	J4KNV2	*Beauveria bassiana*	Complex I intermediate-associated protein	45.9
gene10795	Scaffold6	PHI:7697	A0A0F7A0T2	*Pseudomonas syringae*	Red/far-red light-sensing bacteriophytochrome	38
gene10920	Scaffold6	PHI:494	Q4WPX2	*Aspergillus fumigatus*	Fatty acid oxygenase	45.7
gene11019	Scaffold6	PHI:5043	E7Q5D7	*Saccharomyces cerevisiae*	Hear shock protein	49.5
gene11058	Scaffold6	PHI:9526	A0A384JQ57	*Botrytis cinerea*	Pectin esterase	48
gene11108	Scaffold6	PHI:6751	Q4WQ36	*Aspergillus fumigatus*	Initiates asexual development	30.7
gene11239	Scaffold6	PHI:8509	G9MQ09	*Trichoderma virens*	Non-ribosomal peptide synthetase	50.7
gene11541	Scaffold6	PHI:2393	I1R980	*Fusarium graminearum*	-	36.8
gene11622	Scaffold6	PHI:7685	A0A0J9UIM0	*Fusarium oxysporum*	Nitrogen status-sensing regulatory protein	53.1
gene11701	Scaffold6	PHI:8896	A0A1D8PKM5	*Candida albicans*	Phosphatidyl-N-methylethanolamine N-methyltransferase	59.5
gene11796	Scaffold6	PHI:2383	Q2VF46	*Monilinia fructicola*	Cutinase	35.6
gene11972	Scaffold6	PHI:3417	W5ZQ93	*Beauveria bassiana*	Mitochondrial transmembrane protein	36.3
gene12319	Scaffold7	PHI:1134	B0LLU1	*Leptosphaeria maculans*	Unknown	41.2
gene12442	Scaffold7	PHI:9526	A0A384JQ57	*Botrytis cinerea*	Pectin esterase	47.6
gene12718	Scaffold7	PHI:8556	O52658	*Pseudomonas aeruginosa*	Acyl carrier protein	37.8
gene13268	Scaffold7	PHI:6090	A0A0R1BRS8	*Acinetobacter nosocomialis*	Hydroperoxidase	45.8
gene13405	Scaffold7	PHI:2393	I1R980	*Fusarium graminearum*	-	34.5
gene13409	Scaffold7	PHI:2393	I1R980	*Fusarium graminearum*	-	24.2
gene13412	Scaffold7	PHI:6956	A0A0J9WWZ9	*Xanthomonas oryzae*	3-oxoacyl-ACP reductase	25.5

## Data Availability

Not applicable.

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
