# Peer review of "Identification, Culture Characteristics and Whole-Genome Analysis of Pestalotiopsis neglecta Causing Black Spot Blight of Pinus sylvestris var. mongolica"

_jof, 2023, doi:10.3390/jof9050564_

Round 1

Reviewer 1 Report

The paper "Identification, Cultural Characteristics and Whole-Genome Analysis of Pestalotiopsis neglecta Causing Black Spot Blight of Pinus sylvestris var. mongolica" is an another example of genome paper, metodically sound and clear description of the genome sequencing of the pathogen. 
The sharp drop in sequencing cost, and availability of genome annotation pipelines allowed to sequence and annotate genomes on routine basis, and this type of paper now can be viewed as "technical reports" rather than real scientific papers. I would like to see more additional analysis and interpretation of the obtained results in a broader context, but still, if the editorial policy allowes to publish this type of paper, I don't mind. 

Speaking of the given paper, I have few remarks:
1. You mentioned that you've assemled reads using HGAP and canu, and polished using Pilon. Please provide versions of the tools used - they may perform VERY different depending on version. Also, provide proper citations (I can't see it for Pilon, for example)
2. " low-quality data can be removed to form clean data" - what was the parameters and thresholds. Information provided in Methods section is insufficient for reproduction of your analysis.
3. Please double check that you are working with the Pestalotiopsis neglecta. I tried to blast sequence that you provided for taxonomic identification (OQ691957), and obtained more than 20 hits with 100% query coverage and 100% identity, including P. lawsoniae, P.lespedezae and P. lushanensis. It is obvious that provided ITS sequence alone is not enough to distinguish these species. Also, Figure 1 does not make any sense - you can take any pair of the sequences from this tree and see that they are identical, or have just 1 nucleotide difference. Thus, you need some additional method for species verification, or reconsider the taxonomy of the Pestalotiopsis genus.
4. "Discussion" and "Conclustions" basically repeats "Results" part. You can make it more short and clear, instead of repeating yoursels.  

Author Response

Dear Reviewer,

I am very grateful to you for giving me the opportunity to modify the manuscript.

Thank you for the comments, and we are very appreciative of the valuable comments from you. Based on these comments and suggestions, we have made careful modifications to the original manuscript.

All modifications in the revised manuscript were marked in red and highlighted in yellow. The point-by-point responses to your comments are listed in the attached document.

We appreciate your warm work earnestly and hope these will make it more acceptable for publication.

If there are further issues to be clarified, please contact us without hesitation. Thank you very much again.

Yours sincerely,

Jing Yang

Reviewer 2 Report

The abstract is improvable. It detail in excess methodological characteristics, but not mention even, nothing about other aspects mentioned in the title. 

The kit used por genomic DNA extraction, according to mention in section 2.3.1, is not specific for fungi, not even recommend; so in the mentioned by authors "manufacterer's protocol" there is not any instruction rergarding its use on fung. The authors must precise the adaptations realized and detail specifically the used protocol.

Nothing is said about a crucial point as is the depth of sequencing. Absolutely relevant in order to establish the confidence of the posterior de novo assembly. This must be corrected.

It is not detailed the algorithms, pipelines or tools used for the bioinformatic processing. Neither primary nor subsequents.  Even when is, only, cited the process used for the genome assembly (HGAP), is not specified version or any other relevant details involved in the required workflow as pre-assembling, trimming etc... This section must be reformuled completely and in function of this could appear deeper objections regarding the suitability of the research design.

The parameters of configuration of the very few bioinformatics tools cited (2.3.2 & 2.3.3) are omitted (error correction, assembly filtering, annotation, etc...). In other cases,  exposing already the Results, as in the "pathogenic system analysis" is cited software (Diamond) that it hasn't been mentioned in the Materials and Methods section (2), etc...

Author Response

Dear Reviewer,

I am very grateful to you for giving me the opportunity to modify the manuscript. Thank you for the comments, and we are very appreciative of the valuable comments from you. Based on these comments and suggestions, we have made careful modifications to the original manuscript. All modifications in the revised manuscript were marked in red and highlighted in yellow. The point-by-point responses to your comments are listed as follows. We appreciate your warm work earnestly and hope these will make it more acceptable for publication. If there are further issues to be clarified, please contact us without hesitation. Thank you very much again.

Comment 1: The abstract is improvable. It detail in excess methodological characteristics, but not mention even, nothing about other aspects mentioned in the title.

Response 1: Thanks for your advice and we have rewrite the abstract as follows:

Black spot needle blight is a serious conifer diseases of Pinus sylvestris var. mongolica occurred in Northeast China, which was usually caused by the plant pathogenic fungus Pestalotiopsis neglecta. From the diseased pine needles collected in Honghuaerji, P. neglecta strain YJ-3 was isolated and identified as the phytopathogen and its cultural characteristics were studied. Then, we generated a highly contiguous, 48.36-Mbp genome assembly (N50 = 6.62 Mbp) of P. neglecta strain YJ-3 by combining PacBio RS II Single Molecule Real Time (SMRT) and Illumina HiSeq X Ten sequencing platforms. Results showed that a total of 13,667 protein-coding genes were predicted and annotated using multiple bioinformatics databases. The genome assembly and annotation resource reported here will be useful for the study of fungal infection mechanisms and pathogen-host interaction.

Comment 2: The kit used por genomic DNA extraction, according to mention in section 2.3.1, is not specific for fungi, not even recommend; so in the mentioned by authors "manufacterer's protocol" there is not any instruction regarding its use on fungi. The authors must precise the adaptations realized and detail specifically the used protocol.

Response 2: Thanks for your reminder. We are sorry that we made an error here and caused a misunderstanding. After further verification, we have corrected the kit for extracting fungal DNA as follows:

Genomic DNA was extracted using the Omega Fungal DNA Kit D3390-02 according to the manufacturer’s protocol. Purified genomic DNA was quantified by TBS-380 fluorometer (Turner BioSystems Inc., Sunnyvale, CA). High quality DNA (OD260/280=1.8~2.0, >15 μg) was used to further construct library.

Comment 3: Nothing is said about a crucial point as is the depth of sequencing. Absolutely relevant in order to establish the confidence of the posterior de novo assembly. This must be corrected.

Response 3: Thanks for pointing this out. After contacting the sequencing company, we have rewritten the methods.

Comment 4: It is not detailed the algorithms, pipelines or tools used for the bioinformatic processing. Neither primary nor subsequents.  Even when is, only, cited the process used for the genome assembly (HGAP), is not specified version or any other relevant details involved in the required workflow as pre-assembling, trimming etc... This section must be reformuled completely and in function of this could appear deeper objections regarding the suitability of the research design.

Response 4: Thanks for your question. In the section of 2.3. Whole-genome sequencing and assembly, we have added the versions and citations of the tools used.

Comment 5: The parameters of configuration of the very few bioinformatics tools cited (2.3.2 & 2.3.3) are omitted (error correction, assembly filtering, annotation, etc...). In other cases, exposing already the Results, as in the "pathogenic system analysis" is cited software (Diamond) that it hasn't been mentioned in the Materials and Methods section (2), etc...

Response 5: Thanks for your question. We have rewritten this part in the section of materials and methods.

Round 2

Reviewer 1 Report

Thanks, I'm satisfied with the current version of the paper. I especially appreciate your efforts on the morphological data and additional measures for identification of the YJ-3 as P. neglecta.

Author Response

Thanks for your approval. Hope you well.

Reviewer 2 Report

The corrections done enhance the previous version of your paper in my opinion. Congratulations.

Nevertheless, remains several errors, that must be corrected, like the sequence of the FW primer for ITS1. The correct is ITS1 (5′-TCCGTAGGTGAACCTGCGG-3′).

Respect to the mentioned depth of sequencing, for both plattforms but specially for HiseqX Ten, is "unsual" for excesively high. You could have used for whole genome seq. projects like this, a 10X depth with (in general, subject to particular revision for target specie) good enough results respect to whole-genome coverage, variant discovery power and or  quality of variants. One aspect with a very considerable influence over the costs of seq. that you could consider in your next works.

Author Response

Dear Reviewer,

Thanks for your approval.

We have corrected the sequence of the FW primer for ITS1 in the text. For your suggestions, we would consider them in our next works.

Thanks very much again and hope you well.